# The Cost of Compression: Investigating the Impact of Compression on Parametric Knowledge in Language Models

**Satya Sai Srinath Namburi, Makesh Sreedhar, Srinath Srinivasan, Frederic Sala**

University of Wisconsin - Madison

{sgnamburi, msreedhar, srinivasan32, fsala}@wisc.edu

## Abstract

Compressing large language models (LLMs), often consisting of billions of parameters, provides faster inference, smaller memory footprints, and enables local deployment. Two standard compression techniques are pruning and quantization, with the former eliminating redundant connections in model layers and the latter representing model parameters with fewer bits. The key tradeoff is between the degree of compression and the impact on the quality of the compressed model. Existing research on LLM compression primarily focuses on performance in terms of general metrics like perplexity or downstream task accuracy. More fine-grained metrics, such as those measuring parametric knowledge, remain significantly underexplored. To help bridge this gap, we present a comprehensive analysis across multiple model families (ENCODER, ENCODER-DECODER, and DECODER) using the LAMA and LM-HARNESS benchmarks in order to systematically quantify the effect of commonly employed compression techniques on model performance. A particular focus is on tradeoffs involving parametric knowledge, with the goal of providing practitioners with practical insights to help make informed decisions on compression. We release our codebase[1] to enable further research.

## 1 Introduction

Large language models (LLMs) have demonstrated exceptional performance across diverse tasks. However, their deployment in real-world applications is hindered by their substantial size and the associated costs, even for inference (Schwartz et al., 2020; Strubell et al., 2019). For instance, the LLama-65B model (Touvron et al., 2023), a pioneering open-sourced LLM, uses approximately 130GB of RAM for 16-bit inference. To address this challenge, recent research has

focused on developing novel compression techniques that enable efficient local deployment and inference. Notable examples of such techniques include SparseGPT (Frantar and Alistarh, 2023) and LLM.int8() (Dettmers et al., 2022).

The tradeoff between model compression and quality is typically studied either through general metrics like perplexity (See et al., 2016; Michel et al., 2019) or standardized benchmark task accuracy (Liang et al., 2021; Du et al., 2021) on, e.g., GLUE (Wang et al., 2018). Furthermore, much of the literature studies such tradeoffs for one model or a particular class of models. Unfortunately, as a result, practitioners *do not have access to reliable insights or rules-of-thumb* to ensure they can make an informed decision for compression in their own models. This is because

- Metrics like perplexity are too general, while benchmark prediction metrics are too easy to fool. For example, recent findings suggest that distilled versions of foundational LLMs, known as imitation models, may exhibit stylistic similarities but potentially lack knowledge when compared to the models they seek to imitate (Gudibande et al., 2023).
- Most recent research on compression techniques has primarily focused on DECODER models. The applicability and effectiveness of such techniques for large ENCODER and ENCODER-DECODER models (Chung et al., 2022) has yet to be extensively studied.

These difficulties suggest that there is a need for a more fine-grained understanding of the effects of compression schemes, comparing a variety of model families, compression techniques, and specialized measurements.

We address these challenges, specifically focusing on the preservation of *parametric knowledge*, i.e., knowledge acquired during pretraining, that is stored in model weights. This is particularly

---

crucial for tasks involving reasoning and for specialized applications. Concretely, we examine the impact of different compression schemes on parametric knowledge across multiple model families (ENCODER, ENCODER-DECODER and DECODER) where we apply pruning and quantization approaches and analyze the performance of such techniques on downstream reasoning tasks. To the best of our knowledge, this work represents one of the first large-scale investigations in this direction. Among the crucial observations resulting from this study include:

- Pruning all modules together has the most significant impact on parametric knowledge, compared to pruning specific modules,
- At pruning levels of >50%, the parametric knowledge of all the models declines rapidly,
- Quantizing attention modules has less impact on performance compared to quantizing feed-forward networks for all the models,
- Across all models, structured pruning at the final layer has detrimental effects compared to unstructured pruning.

## 2 Background

In this section, we briefly discuss the various compression techniques we use in our study.

### 2.1 Pruning

Pruning involves reducing the model size by eliminating unnecessary or redundant connections between neurons or entire neurons altogether. Broadly speaking, pruning approaches can be classified into two types (Fig. 1):

**Unstructured Pruning:** Each connection is treated as an individual entity, and sparsity is attained by eliminating connections with lower saliency. Although this approach enables the removal of less important connections without compromising performance, it leads to sparse matrix operations, which may not be optimal for certain hardware accelerators[2] (Buluc and Gilbert, 2008; Gale et al., 2019).

**Structured Pruning:** This involves removing a group of connections, such as channels or entire neurons, instead of individual connections. Unlike unstructured pruning, this approach avoids introducing sparse matrix operations. However, aggres-

---

[2]The current landscape is evolving as advanced accelerators are emerging that provide support for sparse multiplications.

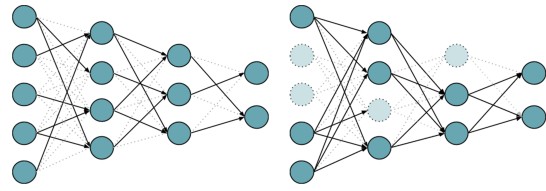

Figure 1: An illustration of unstructured (left) vs structured (right) pruning.

sive structured pruning may disproportionately impact the model's performance (Yao et al., 2019).

**Choosing Saliency of Weights:** When choosing the criterion to determine saliency, various factors can be taken into account, such as weight magnitude, importance to the overall network functionality, or contribution to specific tasks. Typically, the saliency of weights is determined based on their magnitudes when selecting which ones to remove during pruning. A sparsity of k% means that the least salient k% connections are removed.

The most commonly used pruning types are:

1. **L1-Unstructured:** Connections between neurons are eliminated individually, and their saliency is determined by their $L_1$-norm, i.e., the smallest weights are removed.
2. **Lp-Structured:** Connections are eliminated in a structured way, i.e., an entire layer/channel is removed, and saliency is determined by their $L_p$-norm where $p$ is a hyperparameter.

### 2.2 Quantization

Model parameters can be categorized into weights and activations, which are typically represented using 32 bits. Quantization aims to reduce the number of bits used for representing these parameters. A popular choice for this mapping is[3]:

$$Q(r) = \text{Int}(r/S) - Z,$$

where $Q$ is the quantization operator, $r$ is a real-valued input (weight or activation), $S$ is a real-valued scaling factor, and $Z$ is an integer zero-point. An important factor in mapping $r$ to an integer is the scaling factor $S$. This is usually given by

$$S = \frac{\beta - \alpha}{2^b - 1}. \tag{1}$$

---

[3]Uniform quantization maps real values to equally spaced integers

Here $[\alpha, \beta]$ denotes the clipping range and $b$ is the quantization bandwidth. The process of determining the clipping range is known as calibration. Extensive research has been conducted to determine the optimal range to reduce the bit representation while balancing accuracy, computational efficiency, and inference speed (Gholami et al., 2021). In most cases, statistics for weights are precomputed as they remain constant during inference. Often, it may be necessary to fine-tune the quantized model parameters to enhance performance on task-specific datasets. Taking these factors into account, various methods have been proposed (Nagel et al., 2021):

**Post Training Static Quantization (PTSQ):** The clipping range for activations is pre-calculated using a representative dataset, which is a small subset derived from the task-specific dataset. Using this clipping range, the activations are quantized in advance and thus remain static during inference.

**Post Training Dynamic Quantization (PTDQ):** The clipping range is dynamically calculated for each activation during inference. Although this introduces additional computational overhead during inference, it yields improved performance compared to Post Training Static Quantization (PTSQ) as the signal range is exactly calculated for each input.

**Quantization Aware Training (QAT):** The model undergoes a process known as fake-quantization, i.e., during training all the calculations involving forward and backward passes are performed in full-precision. Subsequently, after updating the weight parameters through gradient descent, the weights are quantized to a lower bit. While this approach achieves the highest performance, it requires finetuning the model.

We note that while a huge diversity of often sophisticated and specialized compression methods have been proposed, we focus on a subset of standard approaches. This enables us to seek more general insights on compression tradeoffs.

## 3 Experimental Setup

In this section, we present a comprehensive overview of our experimental setup, including the rationale behind our design choices, along with the selection of models and datasets.

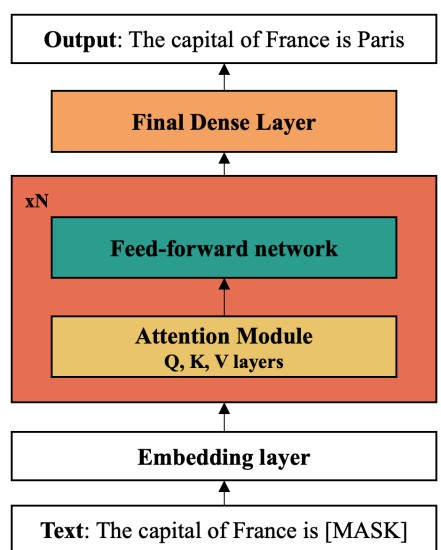

Figure 2: Block diagram of a simplified Transformer describing modules compressed in our experiments.

### 3.1 Settings Under Consideration

The general transformer block consists of an attention module followed by a feed-forward network. As a result, we consider three choices for compression: compress the attention module alone §3.2, compress the feed-forward network alone §3.3, or compress both together §3.4. Figure 2 contains a visual representation of these modules.

Our chosen compression techniques include pruning, quantization, and a combination of pruning and quantization. Following the methodology proposed in Han et al. (2015), we adhere to the sequential order of pruning the selected group of modules first and then applying quantization. In addition, we also investigate the impact on distilled models and explore the effects of employing various combined compression techniques.

### 3.2 Attention-only Global Compression 🟨

We include all the linear layers within all the attention modules of the model. For encoder-decoder models, we also consider the cross-attention blocks.

**Attention-only Global Pruning, ($Att_{GP}$):** We apply pruning to all the linear layers within the attention modules.

**Attention-only Global Quantization, ($Att_{GQ}$):** We quantize all the linear layers within the attention modules.

**Attention-only Global Pruning + Quantization, ($Att_{GPQ}$):** We prune the linear layers in

the attention modules and subsequently quantize them.

### 3.3 Feed-forward-only Global Compression ■

We include all the linear layers within all the feed-forward networks of the model.

**Feed-forward-only Global Pruning, ($FF_{GP}$):** We employ pruning to all the linear layers within the feed-forward networks.

**Feed-forward-only Global Quantization, ($FF_{GQ}$):** We quantize all the linear layers within the feed-forward networks.

**Feed-forward-only Global Pruning + Quantization, ($FF_{GPQ}$):** We prune all the linear layers from feed-forward networks and subsequently quantize them.

### 3.4 Overall Global Compression

We specifically target the linear layers within the attention and feed-forward network. Under this compression, the different setups are:

**Overall Global Pruning, ($Overall_{GP}$) ■:** We employ pruning to all the linear layers (except the final dense layer).

**Overall Global Quantization,($Overall_{GQ}$) ■+■:** We apply quantization to all the linear layers (including the final dense layer).

**Overall Global Pruning ■ + Quantization ($Overall_{GPQ}$) ■+■:** We first apply pruning to all the linear layers (except the final dense layer), and subsequently, we quantize all the linear layers.

### 3.5 Final Dense Layer Pruning, ($FL_P$) ■

Recent studies (Mitchell et al., 2021, 2022; Meng et al., 2022) provide evidence suggesting that the final layers of a language model play a significant role in storing information. Given its importance, we focus on understanding how knowledge is encoded in the final layer. Therefore, we treat the final layer as an individual module in our experimental setup and prune it. We consider L1-structured and L1-unstructured pruning as outlined in §2.1.

We note that the number of parameters compressed differs for different settings. We record all of the values required for normalizing measurements. However, our focus is predominantly aimed at *understanding the effects of compressing modules and their combinations* rather than presenting normalized results, and our insights reflect this framing. We provide full parameter counts

that permit normalized quantities that can be used by practitioners who seek to directly apply our work and refer the readers to Sec. A.2 for more details.

### 3.6 Design Choices

- In our global pruning experiments ($Overall_{GP}$, $Att_{GP}$, $FF_{GP}$), we use L1-Unstructured and apply pruning percentages ranging from 10% to 90% with increments of 10%.
- For quantization experiments, as we seek to investigate the zero-shot capabilities of LLMs, we select post-training dynamic quantization §2.2, eliminating the need for finetuning (unlike quantization-aware training; QAT §2.2) or calibration of the model to a representative dataset (unlike post-training static quantization; PTSQ §2.2) and quantize to 8 bits (int8).
- Since the quantization of activations occurs during inference, which is dynamic in nature, the order of inputs within a batch has a minor impact on the final accuracy ($< 1\%$). Therefore, we seed the experiments to ensure consistent and reproducible results (§A.1).
- Previous studies (Gordon et al., 2020; Michel et al., 2019) suggest that pruning levels of 30%-40% do not affect the model on downstream tasks. Such rules-of-thumb may or may not hold for parametric knowledge. In our experimental settings ($GPQ$, $FL_P$), we select 20% and 40% as the levels to understand when a similar result holds.

### 3.7 Model Zoo

We consider the following models for our study. Where available, we choose both the base and large versions of the model to understand if larger models exhibit different behavior.

#### 3.7.1 Encoder-only:

- **BERT** (Devlin et al., 2019): Pretrained on masked language modeling (MLM) and next sentence prediction (NSP) objective.
- **RoBERTa** (Liu et al., 2019): Similar to BERT with different training choices (larger training dataset and removed NSP).
- **DistilBERT** (Sanh et al., 2020): Distilled version of BERT whose training objective includes MLM, a distillation loss, and a cosine embedding loss.

- **ALBERT** (Lan et al., 2019): Parameter-reduced version of BERT using cross-layer parameter sharing and factorized embedding parameterization.

### 3.7.2 Encoder-Decoder:

- **Flan-T5** (Chung et al., 2022): Instruction-finetuned encoder-decoder model with masked span corruption objective.
- **Lamini-Flan-T5** (Wu et al., 2023): Flan-T5 model finetuned on LaMini instruction dataset[4] which is generated and distilled using ChatGPT output.

### 3.7.3 Decoder only:

- **Vicuna-7B** (Chiang et al., 2023): An instruction-based LLama derived model finetuned on user-shared conversations collected from ShareGPT.
- **WizardLM-7B** (Xu et al., 2023): An instruction-based LLama derived model with instructions generated by LLMs (rather than humans) using the Evol-Instruct mechanism.

### 3.8 Datasets

We use the following datasets for our empirical analysis:

**LAMA:** To examine the effects of compression on encoder-only models, we use the LAMA (LAnguage Model Analysis) benchmark (Petroni et al., 2019). LAMA assesses the factual and commonsense knowledge of language models. Each example in LAMA is formulated as a cloze-style question, where either the subject or object is masked. By predicting the masked word, we can evaluate the model's ability to recover real-world facts. Specifically, we probe the encoder-only models with LAMA to investigate the impact of compression on various knowledge tasks. This benchmark consists of four datasets, namely TRex, Google-RE, ConceptNet, and SQUAD, each designed to assess specific types of relational knowledge. These datasets provide valuable insights into the model's performance and its understanding of different types of information.

**Language model evaluation harness:** To examine the effects of compression on encoder-decoder and decoder-only models, we use a subset of evaluation harness tasks (Gao et al., 2021): the BoolQ dataset (Clark et al., 2019), the PIQA

---

[4]https://huggingface.co/datasets/MBZUAI/LaMini-instruction

dataset (Bisk et al., 2020), and the Winogrande dataset (Sakaguchi et al., 2021). These datasets provide a range of challenging prompts for each model type. We refer the reader to Table 2 for examples of samples from each dataset.

## 4 Experimental Results and Insights

To facilitate our discussion, we categorize pruning levels as follows:

- $p_{low}$: Sparsity levels of 10-30%
- $p_{medium}$: Sparsity levels of 30-50%
- $p_{high}$: Sparsity levels of >50%

For encoder-only models, we report the % drop in top-1 accuracy, averaged across all the probes in LAMA. For the decoder-only and encoder-decoder models, we report the % drop in accuracy, averaged across BoolQ, PIQA and Winogrande. In the decoder-only and encoder-decoder plots, the majority-baseline indicates the accuracy when all the predictions are assigned to the majority class.

### 4.1 Global Pruning

We observe that for encoder-only models (Fig. 3, 19), there is a minimal decline in performance at $p_{low}$. At $p_{medium}$, the drop in performance is more significant for pruning feed-forward networks ($FF_{GP}$) as compared to attention modules ($Att_{GP}$).

> **Finding:** At $p_{medium}$, for encoder-only models, pruning attention modules ($Att_{GP}$) has a smaller impact compared to pruning feed-forward networks ($FF_{GP}$).

We observe that for encoder-decoder (Fig. 5, 13) and decoder-only models (Fig. 4), there is a minimal decline in performance at $p_{low}$. However, at $p_{medium}$, the drop in performance is more significant for pruning attention modules ($Att_{GP}$) compared to feed-forward networks ($FF_{GP}$).

> **Finding:** At $p_{medium}$, for encoder-decoder and decoder-only models, pruning the attention module ($Att_{GP}$) has more impact on performance compared to pruning feed-forward networks ($FF_{GP}$).

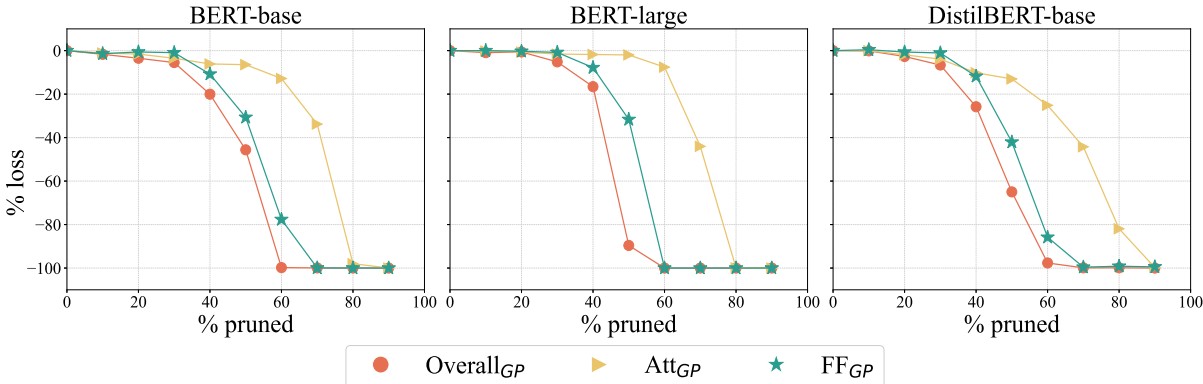

Figure 3: Averaged drop in Top-1 accuracy for encoder-only models for global pruning.

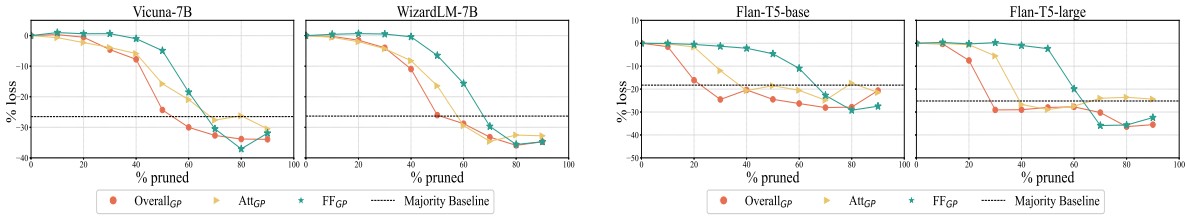

Figure 4: Averaged drop in accuracy for decoder-only models for global pruning.

Figure 5: Averaged drop in accuracy for encoder-decoder models for global pruning.

We note that the number of parameters in the feed-forward networks is significantly higher than the number of parameters in the attention modules for all these models (Table 3). This observation provides a likely explanation for the pattern observed in encoder-only models, where pruning more parameters results in a higher loss of parametric knowledge. However, *this finding is counterintuitive for encoder-decoder and decoder-only models*, as we would expect that pruning the larger feed-forward networks would have a more significant impact on the parametric knowledge. We suspect that the feed-forward networks are over-parameterized and thus they can be pruned without a significant drop in performance.

> **Finding:** For all the models, pruning all the modules together ($Overall_{GP}$) has the most significant impact on performance.

Among all the models analyzed, pruning all modules together ($Overall_{GP}$) has the most significant negative impact on performance. This finding suggests that when compressing models, pruning all modules simultaneously leads to a greater loss of parametric knowledge compared to pruning specific modules or components individually. Therefore, it is crucial to carefully consider

the implications of employing global pruning techniques. We additionally note that at $p_{high}$, the performance goes to zero as expected.

Additional results for global pruning on individual datasets for encoder-only models are shown in Fig 20, 21, 22; for decoder-only models at Fig 23; for encoder-decoder models at Fig 13, 25.

## 4.2 Global Quantization

We observe that across all the models (Fig. 6, 15, 16), the performance drop is less significant when quantizing attention modules ($Att_{GQ}$) compared to quantizing feed-forward networks alone ($FF_{GQ}$). This contrasts with the results from global pruning (§4.1), where pruning attention-only modules had a more detrimental effect on encoder-decoder and decoder-only models.

> **Finding:** For all the models, quantizing attention modules ($Att_{GQ}$) has lesser impact compared to quantizing feed-forward networks ($FF_{GQ}$).

We hypothesize that in the case of quantization where all connections are preserved, the parametric knowledge in cross-attention modules may remain *relatively intact*. However, in pruning, as connections are eliminated, there may have

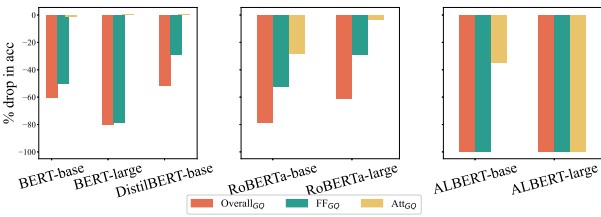

Figure 6: Averaged drop in Top-1 accuracy for encoder-only models for global quantization.

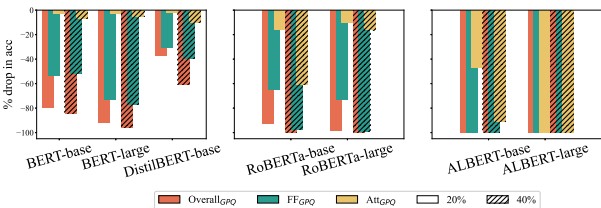

Figure 7: Averaged drop in Top-1 accuracy for encoder-only models for global pruning+quantization.

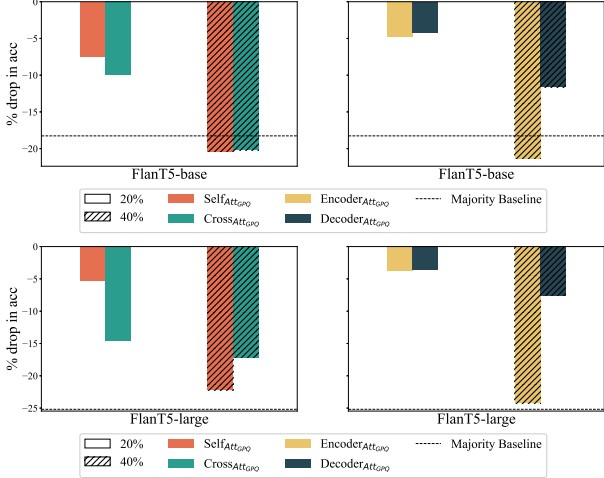

Figure 8: Averaged drop in accuracy for encoder-decoder models for different attention modules compression. $Self_{Att_{GPQ}}$: Compressing only self-attention modules, $Cross_{Att_{GPQ}}$: Compressing only cross-attention modules, $Encoder_{Att_{GPQ}}$: Compressing attention modules in encoder only, $Decoder_{Att_{GPQ}}$: Compressing attention modules in decoder only.

a greater impact on the parametric knowledge in cross-attention modules, thereby affecting the overall capabilities of the model. It is also interesting to observe that the performance drop during quantization is almost similar to that of $p_{medium}$.

> **Finding:** For all the models, quantizing all the modules together ($Overall_{GQ}$) hurts the most.

It is intuitive that quantizing all the modules together ($Overall_{GQ}$) has the most significant negative impact. Additional results are shown in Tables. 4, 5, 6.

### 4.3 Global Pruning + Quantization

For all the models (Fig. 7, 17, 18), at 20% sparsity, compressing attention modules ($Att_{GPQ}$) results in a smaller performance drop compared to compressing feed-forward networks ($FF_{GPQ}$). At 40% sparsity, the same trend is observed for encoder-only and decoder-only models. However, we notice the reverse for ENCODER-DECODER models i.e., that compressing feed-forward networks affects performance *less* than compressing the attention modules at 40% sparsity.

> **Finding:** For all the models, at 20% sparsity level, $Att_{GPQ}$ hurts less compared to $FF_{GPQ}$.

We hypothesize that the sequential effects of

pruning and quantization on the cross-attention modules could be responsible for this change in the order of impact. To test this hypothesis, we selectively prune and quantize the self-attention and cross-attention modules separately and found out that it is indeed the case (Fig. 8) and aligns with the claim made in Michel et al. (2019). Additional results for compressing attention-only modules are shown in Fig 12, 24. For fine-grained analysis on individual datasets, we refer the interested reader to Table 4, 5, 6.

### 4.4 Final Dense layer Pruning

For encoder-only models (Fig. 9), L1-unstructured pruning has a smaller impact compared to L1-structured pruning. We hypothesize that the final layer of the encoder-only models might encode knowledge in a structured or modular manner, and *any form of structured compression would disrupt this encoding*, resulting in a larger performance drop. Such a result would be consistent with existing approaches that enable editing knowledge in language models and rely on structure (Mitchell et al., 2021).

> **Finding:** For encoder-only models, L1-unstructured leads to a smaller decrease in performance than L1-structured.

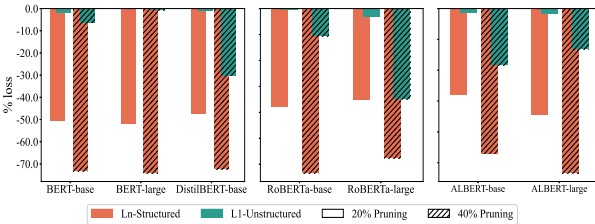

Figure 9: Averaged drop in Top-1 accuracy for encoder-only models for final layer pruning.

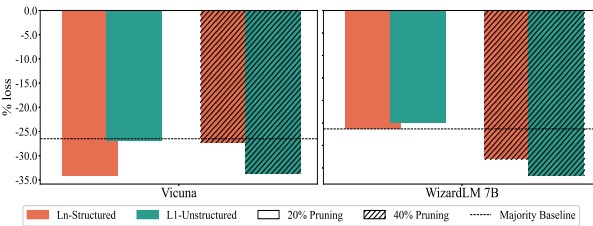

Figure 10: Averaged drop in Top-1 accuracy for decoder-only models for final layer pruning.

For decoder-only (Fig. 10) and encoder-decoder (Fig. 14) models, even at a sparsity level of 20%, the predicted accuracy is very close to the majority baseline. This finding aligns with the claims made in Mitchell et al. (2022) that final layers encode significant amount of information. The drastic performance drop observed suggests that the final layers play a crucial role in encoding knowledge. Additional results for pruning the final layer are shown in Fig. 26, 27, 28.

## 5 Related Work

Early works seeking to understand large language model behavior focused on contextual representations and how such models gain linguistic capabilities (Goldberg, 2019; Ettinger et al., 2018; Jawahar et al., 2019). More recently, some lines of work have steered towards understanding how these models acquire factual and commonsense knowledge. Techniques such as probing evolved as a way to understand the knowledge capabilities of these models (Petroni et al., 2019; Kassner and Schütze, 2020; Talmor et al., 2020; Weir et al., 2020; Wallat et al., 2021).

Previous works including Gordon et al. (2020); Michel et al. (2019) pruned BERT and showed that it is resilient to a medium level of pruning. For example, Michel et al. (2019) showed that after finetuning for a particular downstream task, it is possible to prune about 40% of the attention weights without any loss in performance. A particular fo-

cus has been to understand the importance of the attention mechanism (Voita et al., 2019; Michel et al., 2019) by pruning the heads. In a similar fashion, works such as Zafrir et al. (2019); Bai et al. (2020); Zadeh et al. (2020); Tao et al. (2022); Prato et al. (2019); Frantar et al. (2022); Dettmers et al. (2023) pushed the limits of quantization on language models. Most of these works have focused on one model class or a particular metric.

In another line of work, a variety of approaches (Li and Liang, 2021; Hu et al., 2021; Liu et al., 2021; Lester et al., 2021) focus on alternatives to traditional finetuning of the model due to its scale. In contrast to these works, our paper primarily focuses on the in-built parametric knowledge present in the model. This means we do not finetune and instead seek to understand whether some of the previously described phenomenona are applicable to other models as well.

Also connected to this work are techniques that edit factual knowledge in models. The goal for such works is to avoid retraining or even finetuning models, instead seeking to directly change parameters connected to certain facts (Mitchell et al., 2021, 2022; Meng et al., 2022). However, given our focus on compression, the main theme of our work differs. Nevertheless, it would be interesting to understand the impact of relying on compressed models when using such editing techniques.

## 6 Conclusion

Compression is crucial in deploying and using large language models. Despite its importance, existing empirical studies predominantly rely on generic measurements such as perplexity or standardized benchmark metrics when investigating the effects of compression. These coarse measurements are challenging to interpret. As a result, it is difficult to use them to develop meaningful heuristics for practitioners.

To address these limitations, we provided a large-scale study that focused on fine-grained effects on quantities like parametric knowledge. We studied a variety of compression choices across multiple model families, providing usable insights into what types of compression schemes have the least and most significant impact on models. We hope this work serves as a useful step towards developing users' intuition for rules-of-thumb when selecting appropriate compression techniques in large language models. For future work, we hope

to add additional, more specialized techniques for large language model compression.

## 7 Limitations

Our research has tackled a diverse combination of models, compression schemes, and compression targets within the vast large language model research area. We note that sophisticated and specialized compression techniques tailored to specific objectives for a particular class of models may exhibit distinct behavior compared to the findings presented in this study. Hence, our work does not aim to present an exhaustive set of findings that universally characterize the impact on parametric knowledge across all conceivable models and compression approaches. We believe that our study serves as a valuable starting point, offering a nuanced examination of prevalent methodologies.

We note, additionally, that we do not directly address the tradeoff between wall-clock inference time versus compression. While this is also an important tradeoff, the impact of compression on inference time contains many intricacies that are best treated with a separate large-scale study.

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

# A   Appendix

The appendix contains all of the results we could not include in the body of the paper. We first discuss the statistical approach of the experiments and the performance drop against compression ratio. Then, we show individual plots for a set of experiments that track decrease in accuracy for several types of compression and models. Next, we provide a table that contains information on the datasets used in our experiments. Afterwards, we provide tables with model details, including parameter counts, and explicit results for compression results across model families. Afterwards, we present a large-scale comparison across datasets for encoder-decoder models under various attention module compression approaches. We provide LAMA probe results and finally, change-in-accuracy plots for a variety of datasets for different model classes.

## A.1   Experimental Approach

Our experiments fall into two categories: deterministic and stochastic. Our experiments on pruning are deterministic as we used L1-unstructured pruning. On the other hand, our quantization experiments have an element of randomness. This is due to our use of PTDQ, which computes a dynamic clipping range. We deliberately struck a balance between the number of trials per setting and the overall number of settings studied. Consequently, we ran experiments with multiple seeds and recorded confidence intervals, as demonstrated in Table 1.

## A.2   Performance drop against compression ratio

Normalizing the x-axis to account for the parameter ratio results in the same plots, but with a skewed x-axis (Fig. 11). Given a specific performance drop percentage, it is highly likely that we can achieve greater parameter compression by targeting feedforward modules rather than attention modules. It is worth noting that across all the models studied, feedforward modules have more parameters than attention module

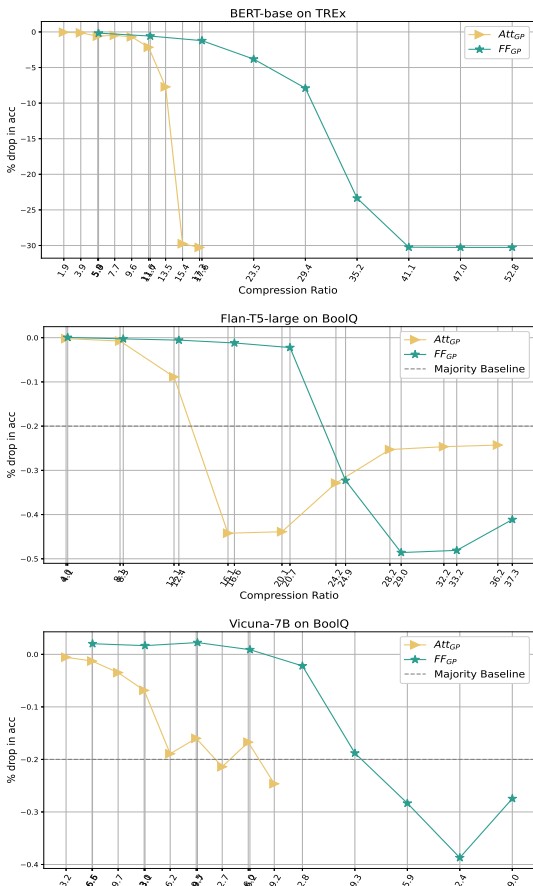

Figure 11: Performance drop vs compression ratios for different model families. Models and Datasets are randomly chosen for representation

Table 1: Top-1 Accuracy from quantizing BERT on SQUAD. Left: BERT-Base, Right: BERT-Large. Baseline for BERT-Base is 12.987 and BERT-Large is 15.909

| | $Overall_{GQ}$ | $Att_{GQ}$ | $FF_{GQ}$ | | $Overall_{GQ}$ | $Att_{GQ}$ | $FF_{GQ}$ |
|---|---|---|---|---|---|---|---|
| Seed 40 | 5.844 | 13.312 | 3.896 | Seed 40 | 2.273 | 16.883 | 2.597 |
| Seed 50 | 5.844 | 13.312 | 5.195 | Seed 50 | 2.922 | 14.286 | 3.896 |
| Seed 60 | 6.169 | 12.987 | 4.221 | Seed 60 | 2.922 | 15.909 | 4.545 |
| **Mean** | 5.952 | 13.204 | 4.437 | **Mean** | 2.706 | 15.693 | 3.679 |
| **Standard Deviation (sample)** | 0.188 | 0.188 | 0.676 | **Standard Deviation (sample)** | 0.375 | 1.312 | 0.992 |
| **Standard Deviation (population)** | 0.153 | 0.153 | 0.676 | **Standard Deviation (population)** | 0.306 | 1.071 | 0.992 |

Table 2: Datasets in our experiments (we use dev sets for BoolQ, PIQA, and Winogrande)

| Probe | Type | #Egs | Question | Answer |
|---|---|---|---|---|
| TRex | Factual | 34k | Francesco Bartolomeo Conti was born in [MASK]. | Florence |
| Google-RE | Factual | 5.5k | Mareva is a [MASK] actress & former beauty Queen | French Polynesia |
| Squad | Factual | 305 | Newton played a [MASK] during Super Bowl 50. | Quarterback |
| ConceptNet | Commonsense | 11k | Joke would make you want to [MASK]. | laugh |
| BoolQ | Mix | 3.2k | Is there any dollar bill higher than a 100? | No |
| PIQA | Commonsense | 1.8k | **Goal:** "ice box" 
 **Soln1:** will turn into a cooler if you add water to it 
 **Soln2:** will turn into a cooler if you add soda to it | Soln1 |
| Winogrande | Commonsense | 1.2k | The trophy doesnt fit into the brown suitcase because *its* too small. 
 The trophy doesnt fit into the brown suitcase because *its* too large. | suitcase 
 trophy |

Table 3: Number of parameters (in million) across all the models

| Model Name | $Overall_{GP}$ | $Att_{GP,GQ,GPQ}$ | $FF_{GP,GQ,GPQ}$ | $FL_P$ | Total trainable parameters (for $Overall_{GQ,GPQ}$) |
|---|---|---|---|---|---|
| Bert-base | 86 | 21 | 64 | 23 | 109 |
| Bert-large | 303 | 75 | 226 | 31 | 334 |
| Roberta-base | 86 | 21 | 64 | 39 | 124 |
| Roberta-large | 303 | 75 | 226 | 51 | 355 |
| Distilbert-base | 43 | 14 | 28 | 23 | 66 |
| Albert-base-v2 | 85 | 28 | 57 | 4 | 89 |
| Albert-large-v2 | 302 | 101 | 201 | 4 | 306 |
| FlanT5-base | 198 | 85 | 113 | 25 | 223 |
| Distil-FlanT5-base | 198 | 85 | 113 | 25 | 223 |
| FlanT5-large | 717 | 302 | 311 | 33 | 750 |
| Distil-FlanT5-large | 717 | 302 | 311 | 33 | 750 |
| Vicuna | 6476 | 2147 | 4329 | 131 | 6607 |
| Wizard-LM | 6476 | 2147 | 4329 | 131 | 6607 |

Table 4: Results from compressing different modules for encoder-only models (numbers represent top-1 accuracy)

| Model | Dataset | Baseline | $Overall_{GQ}$ | $Att_{GQ}$ | $FF_{GQ}$ | $Overall_{GPQ}$ 20% | $Overall_{GPQ}$ 40% | $Att_{GPQ}$ 20% | $Att_{GPQ}$ 40% | $FF_{GPQ}$ 20% | $FF_{GPQ}$ 40% |
|---|---|---|---|---|---|---|---|---|---|---|---|
| **BERT-base** | TREx | 30.27 | 11.536 | 29.903 | 16.614 | 5.552 | 4.233 | 29.675 | 29.217 | 14.691 | 14.628 |
| | GoogleRe | 10.29 | 4.374 | 10.109 | 5.662 | 2.377 | 1.96 | 9.873 | 9.673 | 4.628 | 5.753 |
| | Squad | 12.987 | 5.844 | 13.312 | 3.896 | 2.597 | 0.974 | 12.338 | 10.39 | 4.87 | 4.87 |
| | Conceptnet | 16.33 | 6.02 | 15.897 | 8.677 | 3.919 | 3.69 | 16.224 | 15.553 | 8.262 | 8.324 |
| **BERT-large** | TREx | 30.485 | 4.376 | 30.539 | 4.592 | 1.277 | 0.848 | 29.624 | 29.195 | 8.145 | 6.673 |
| | GoogleRe | 10.472 | 3.829 | 10.309 | 3.612 | 1.434 | 0.436 | 10.018 | 9.964 | 4.247 | 3.376 |
| | Squad | 15.909 | 2.273 | 16.883 | 2.597 | 1.948 | 0.325 | 15.584 | 14.935 | 3.247 | 2.597 |
| | Conceptnet | 19.534 | 4.652 | 19.048 | 5.649 | 1.818 | 1.342 | 18.801 | 18.086 | 5.164 | 4.864 |
| **RoBERTa-base** | TREx | 11.9 | 3.566 | 8.014 | 8.421 | 1.424 | 0.006 | 11.529 | 4.663 | 6.489 | 0.048 |
| | GoogleRe | 4.102 | 1.143 | 2.668 | 1.234 | 0.617 | 0 | 3.249 | 1.779 | 1.053 | 0.091 |
| | Squad | 8.442 | 0.325 | 4.545 | 2.922 | 0 | 0 | 5.195 | 1.623 | 1.299 | 0 |
| | Conceptnet | 17.036 | 3.769 | 14.467 | 7.247 | 0.83 | 0.026 | 14.865 | 8.147 | 5.746 | 0.856 |
| **RoBERTa-large** | TREx | 16.862 | 6.264 | 16.159 | 11.885 | 0.175 | 0.019 | 15.845 | 15.714 | 5.355 | 0.057 |
| | GoogleRe | 3.811 | 1.488 | 3.829 | 2.868 | 0.036 | 0 | 2.196 | 1.869 | 0.926 | 0.054 |
| | Squad | 13.636 | 4.87 | 13.312 | 7.468 | 0 | 0 | 12.013 | 10.714 | 1.299 | 0 |
| | Conceptnet | 19.861 | 8.324 | 19.119 | 15.244 | 1.006 | 0.079 | 18.775 | 17.036 | 7.15 | 0.494 |
| **DistilBERT-base** | TREx | 28.082 | 14.186 | 28.184 | 20.383 | 18.285 | 12.673 | 27.021 | 25.229 | 20.367 | 18.593 |
| | GoogleRe | 10.181 | 4.791 | 10.073 | 8.766 | 5.681 | 3.92 | 9.111 | 8.403 | 8.113 | 7.241 |
| | Squad | 10.39 | 5.195 | 10.714 | 6.494 | 6.494 | 2.273 | 11.688 | 9.74 | 5.519 | 5.195 |
| | Conceptnet | 14.308 | 6.391 | 14.132 | 9.101 | 9.339 | 5.976 | 14.105 | 13.346 | 9.727 | 7.238 |
| **ALBERT-base** | TREx | 13.016 | 0 | 9.213 | 0.003 | 0 | 0.003 | 7.595 | 0.845 | 0 | 0.003 |
| | GoogleRe | 1.307 | 0 | 0.762 | 0 | 0 | 0 | 0.436 | 0.036 | 0 | 0 |
| | Squad | 3.896 | 0 | 1.623 | 0 | 0 | 0 | 1.299 | 0.649 | 0 | 0 |
| | Conceptnet | 9.86 | 0.018 | 6.682 | 0.009 | 0 | 0 | 5.517 | 1.077 | 0.009 | 0.018 |
| **ALBERT-large** | TREx | 22.057 | 0 | 0 | 0.133 | 0 | 0.013 | 0.003 | 0.006 | 0 | 0.003 |
| | GoogleRe | 2.686 | 0 | 0 | 0 | 0 | 0 | 0 | 0 | 0.018 | 0 |
| | Squad | 9.74 | 0 | 0 | 0 | 0 | 0 | 0 | 0 | 0 | 0 |
| | Conceptnet | 14.794 | 0.009 | 0.071 | 0.132 | 0 | 0.009 | 0.026 | 0.044 | 0 | 0.009 |

Table 5: Results from compressing different modules of decoder-only models (numbers represent accuracy). Majority baselines are - **BoolQ**: 0.621, **PIQA**: 0.504, **Winogrande**: 0.504

| Model | Dataset | Baseline | $Overall_{GQ}$ | $Att_{GQ}$ | $FF_{GQ}$ | $Overall_{GPQ}$ | | $Att_{GPQ}$ | | $FF_{GPQ}$ | |
|---|---|---|---|---|---|---|---|---|---|---|---|
| | | | | | | 20% | 40% | 20% | 40% | 20% | 40% |
| **Vicuna-7B** | Boolq | 0.7657 | 0.5211 | 0.7645 | 0.5437 | 0.5272 | 0.4398 | 0.7125 | 0.556 | 0.5346 | 0.4495 |
| | Piqa | 0.778 | 0.611 | 0.7671 | 0.6556 | 0.5979 | 0.5332 | 0.7617 | 0.7421 | 0.6202 | 0.6257 |
| | Winogrande | 0.6725 | 0.5391 | 0.678 | 0.5825 | 0.5043 | 0.4925 | 0.663 | 0.6298 | 0.5612 | 0.5367 |
| **WizardLM-7B** | Boolq | 0.7844 | 0.6073 | 0.7841 | 0.6003 | 0.5817 | 0.4453 | 0.7514 | 0.6801 | 0.6141 | 0.5547 |
| | Piqa | 0.7622 | 0.6518 | 0.7508 | 0.6654 | 0.623 | 0.5593 | 0.7481 | 0.728 | 0.6556 | 0.6371 |
| | Winogrande | 0.6646 | 0.5517 | 0.6638 | 0.588 | 0.5312 | 0.5193 | 0.6622 | 0.6346 | 0.5738 | 0.5817 |

Table 6: Results from compressing different modules of encoder-decoder models (numbers represent accuracy). Majority baselines are - **BoolQ**: 0.621, **PIQA**: 0.504, **Winogrande**: 0.504

| Model | Dataset | Baseline | $Overall_{GQ}$ | $Att_{GQ}$ | $FF_{GQ}$ | $Overall_{GPQ}$ | | $Att_{GPQ}$ | | $FF_{GPQ}$ | |
|---|---|---|---|---|---|---|---|---|---|---|---|
| | | | | | | 20% | 40% | 20% | 40% | 20% | 40% |
| **FlanT5-Base** | Boolq | 0.7887 | 0.618 | 0.7841 | 0.6352 | 0.5049 | 0.482 | 0.7609 | 0.5058 | 0.6275 | 0.6125 |
| | Piqa | 0.6621 | 0.6251 | 0.6665 | 0.6415 | 0.5724 | 0.5419 | 0.6605 | 0.5631 | 0.6393 | 0.6077 |
| | Winogrande | 0.5422 | 0.4862 | 0.5359 | 0.5138 | 0.5051 | 0.4949 | 0.5272 | 0.5241 | 0.5075 | 0.498 |
| **FlanT5-Large** | Boolq | 0.8645 | 0.8034 | 0.8615 | 0.8165 | 0.6498 | 0.5618 | 0.856 | 0.4107 | 0.819 | 0.7969 |
| | Piqa | 0.7138 | 0.6638 | 0.716 | 0.6942 | 0.6181 | 0.5555 | 0.7214 | 0.6616 | 0.6953 | 0.6921 |
| | Winogrande | 0.5991 | 0.5375 | 0.5896 | 0.573 | 0.5185 | 0.5067 | 0.5864 | 0.4957 | 0.5596 | 0.5572 |
| **Lamini-Flan-T5-248M** | Boolq | 0.7982 | 0.7297 | 0.8015 | 0.7346 | 0.667 | 0.4349 | 0.7569 | 0.6266 | 0.7315 | 0.7263 |
| | Piqa | 0.6676 | 0.6393 | 0.6594 | 0.6507 | 0.6208 | 0.5462 | 0.6627 | 0.6208 | 0.6534 | 0.6338 |
| | Winogrande | 0.5304 | 0.5257 | 0.5083 | 0.513 | 0.543 | 0.5051 | 0.5099 | 0.5004 | 0.4964 | 0.5028 |
| **Lamini-Flan-T5-783M** | Boolq | 0.8306 | 0.7982 | 0.8294 | 0.7979 | 0.7716 | 0.6211 | 0.8226 | 0.6783 | 0.7994 | 0.7982 |
| | Piqa | 0.7073 | 0.673 | 0.7051 | 0.6899 | 0.6855 | 0.6192 | 0.7008 | 0.6937 | 0.6882 | 0.6866 |
| | Winogrande | 0.5549 | 0.5241 | 0.5454 | 0.5517 | 0.5193 | 0.4878 | 0.5478 | 0.5114 | 0.5288 | 0.5201 |

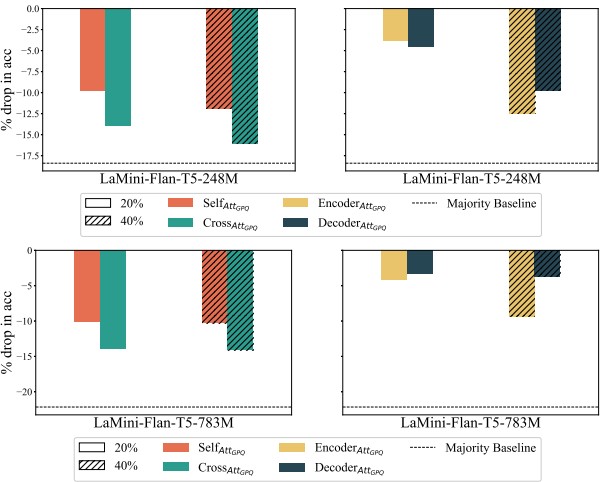

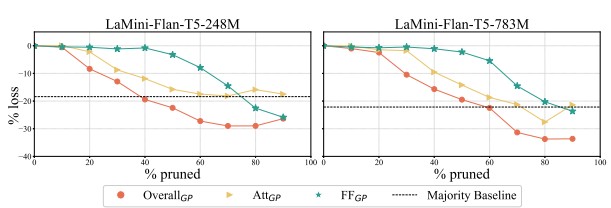

Figure 12: Averaged drop in accuracy for Lamini models under various attention modules compression (§4.3)

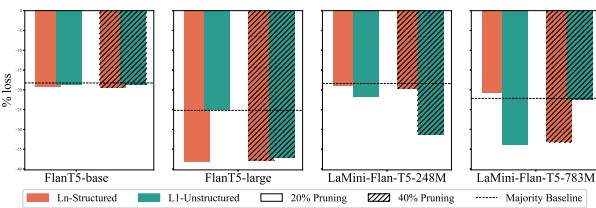

Figure 14: Averaged drop in accuracy for encoder-decoder models for various local pruning (§4.4)

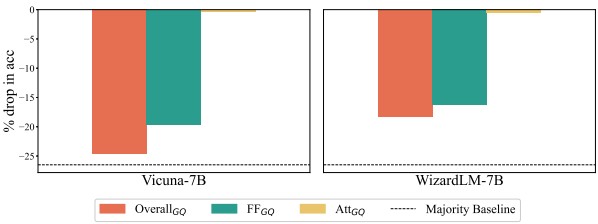

Figure 15: Averaged drop in accuracy for global quantization for decoder-only models (§4.2)

Figure 13: Averaged drop in accuracy for global pruning (§4.1)

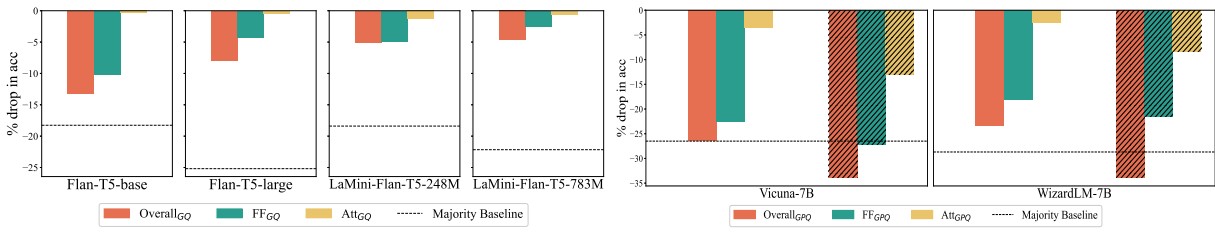

Figure 16: Averaged drop in accuracy for global quantization for encoder-decoder models (§4.2).

Figure 17: Averaged drop in accuracy for global pruning+quantization for decoder-only models (§4.3).

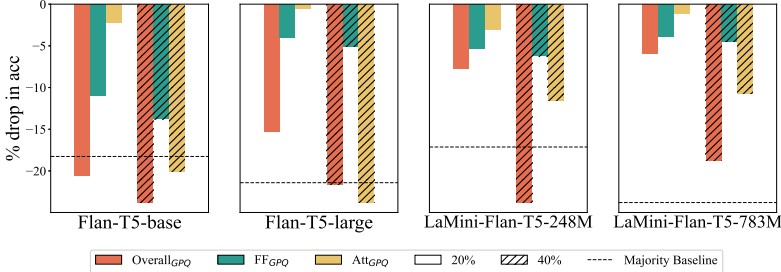

Figure 18: Averaged drop in accuracy for global pruning+quantization for encoder-decoder models (§4.3).

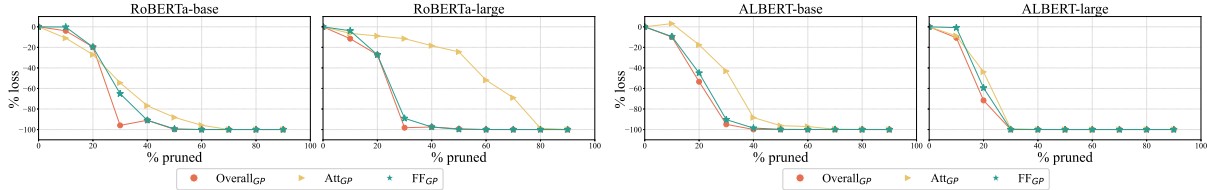

Figure 19: Averaged drop in Top-1 accuracy for encoder-only models (§4.1).

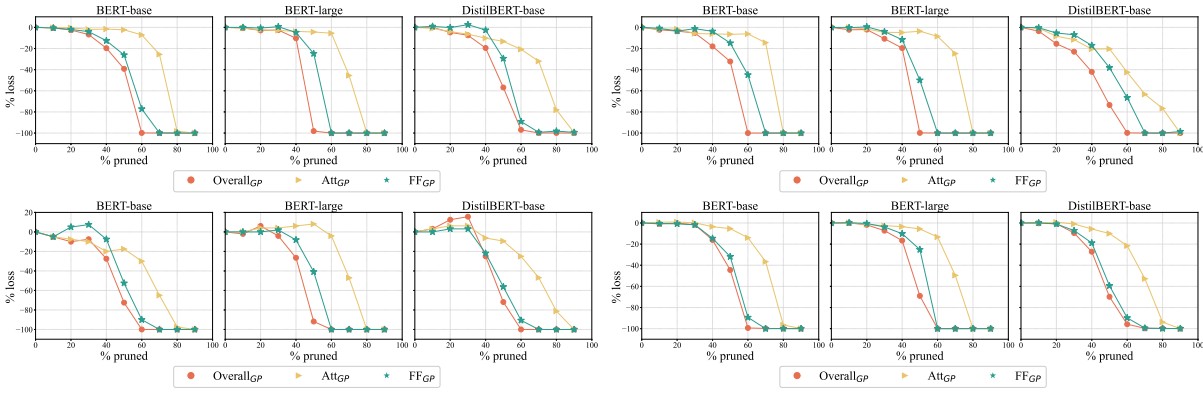

Figure 20: Drop in Top-1 accuracy for respective LAMA probes. Left-to-Right, Top-to-bottom: TREx, Google-RE, SQUAD, Conceptnet (§4.1).

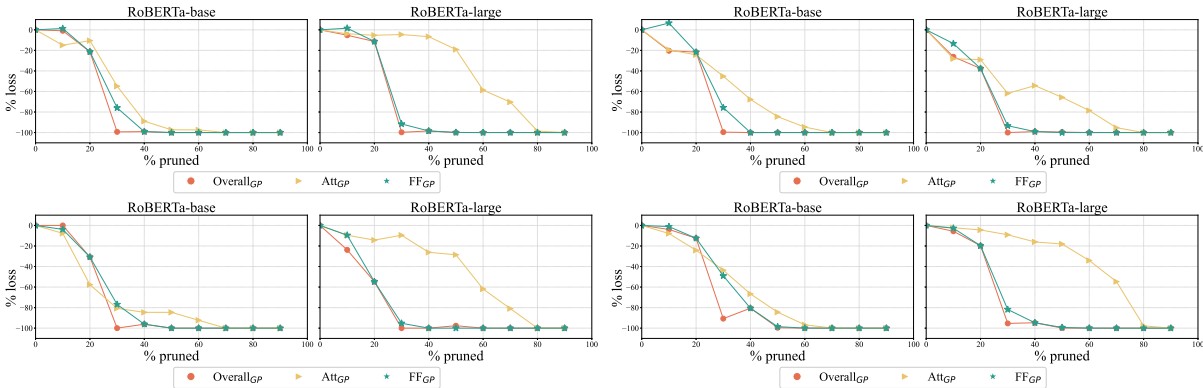

Figure 21: Drop in Top-1 accuracy for respective LAMA probes. Left-to-Right, Top-to-bottom: TREx, Google-RE, SQUAD, Conceptnet (§4.1).

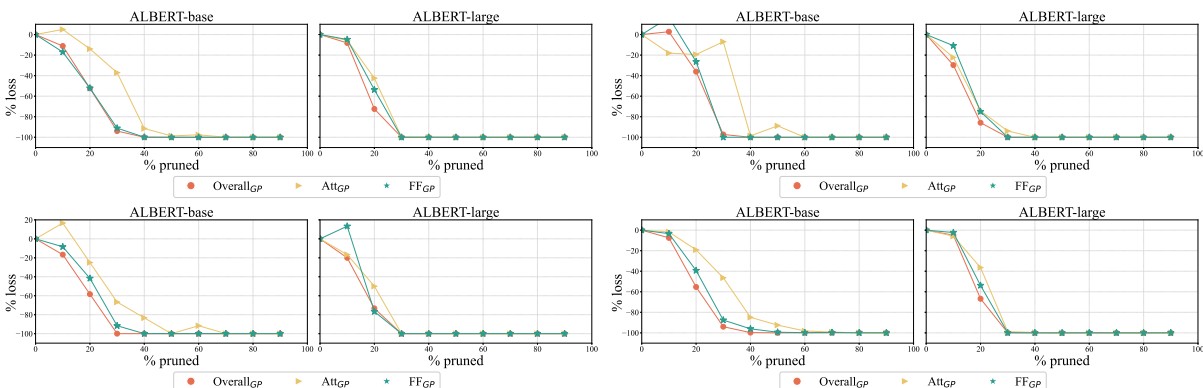

Figure 22: Drop in Top-1 accuracy for respective LAMA probes. Left-to-Right, Top-to-bottom: TREx, Google-RE, SQUAD, Conceptnet (§4.1).

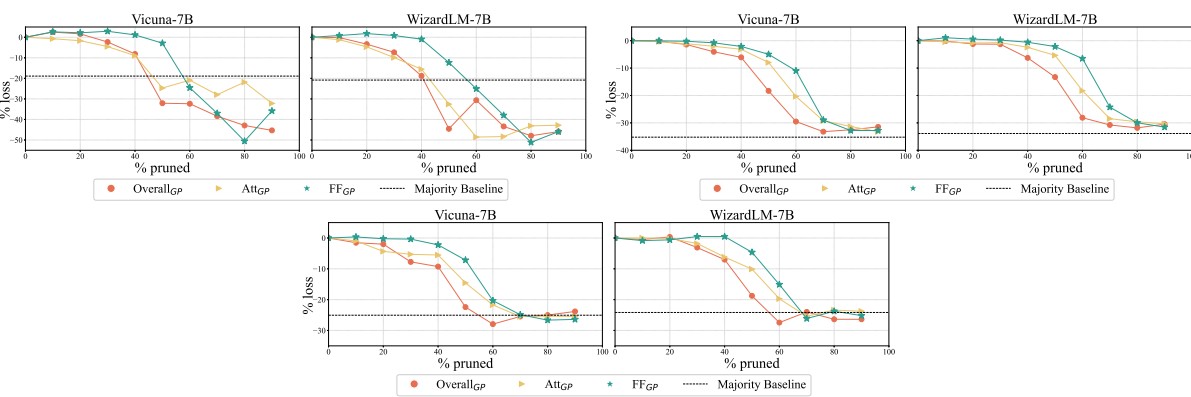

Figure 23: Drop in accuracy for decoder-only models. Left-to-Right, Top-to-Bottom: BoolQ, PIQA, and Wino-grande (§4.1)

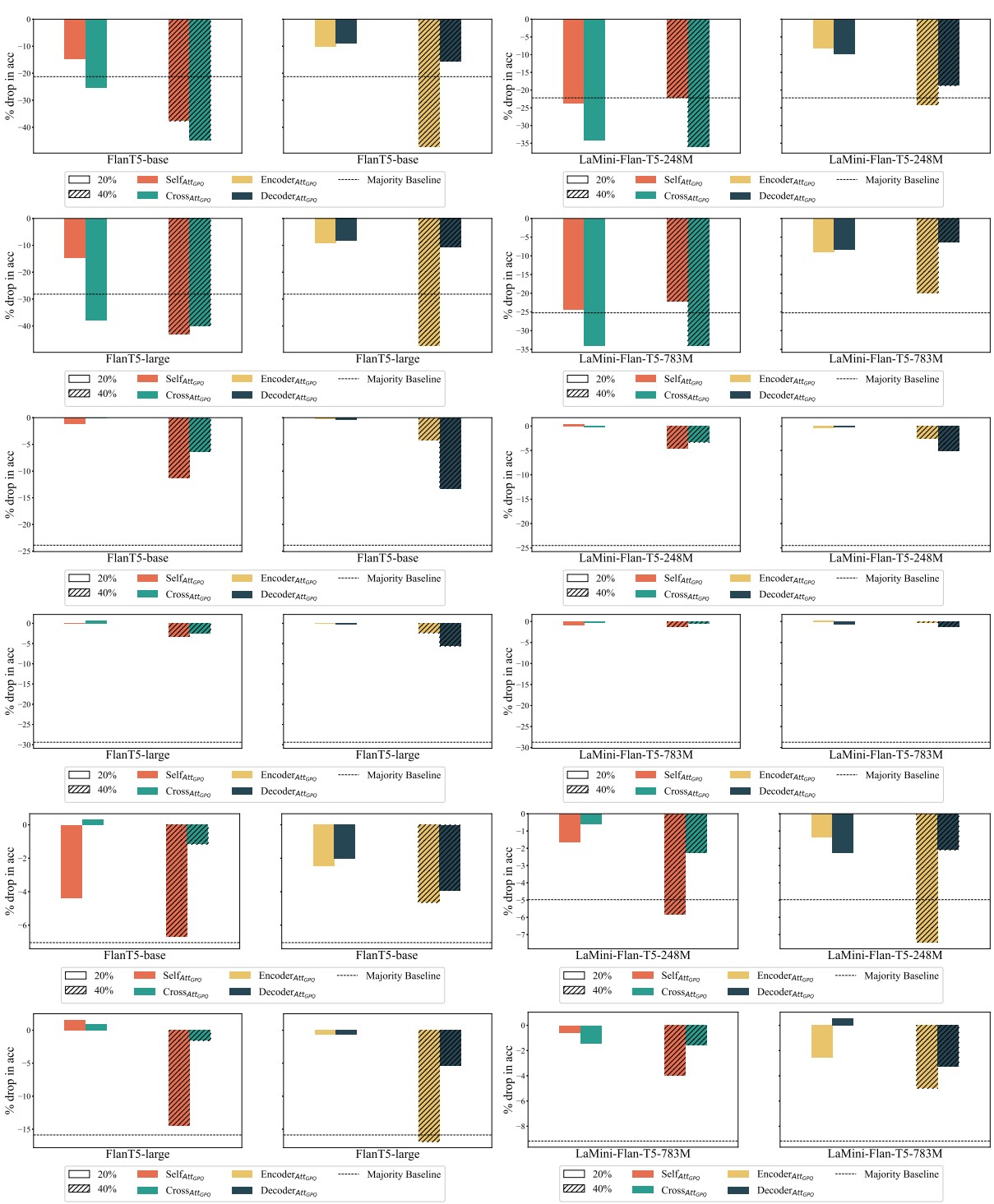

Figure 24: Drop in accuracy across various datasets for encoder-decoder models under various attention modules compression. Top-to-Bottom: BoolQ, PIQA, Winogrande (§4.3)

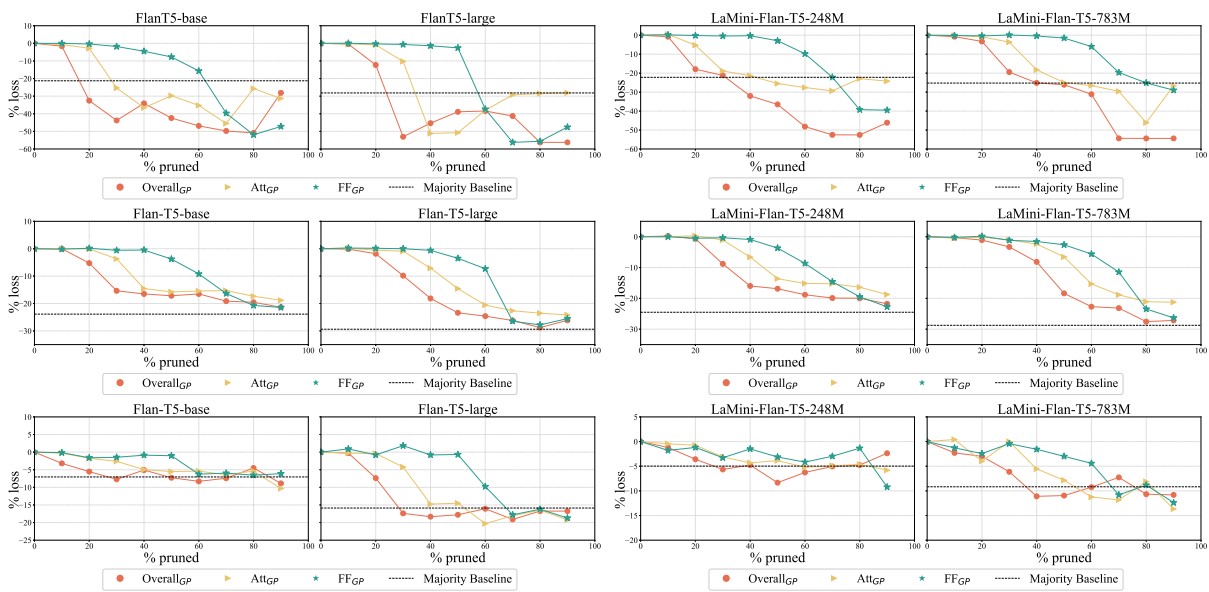

Figure 25: Drop in accuracy across various datasets for encoder-decoder models. Top-to-Bottom: BoolQ, PIQA, Winogrande (§4.1)

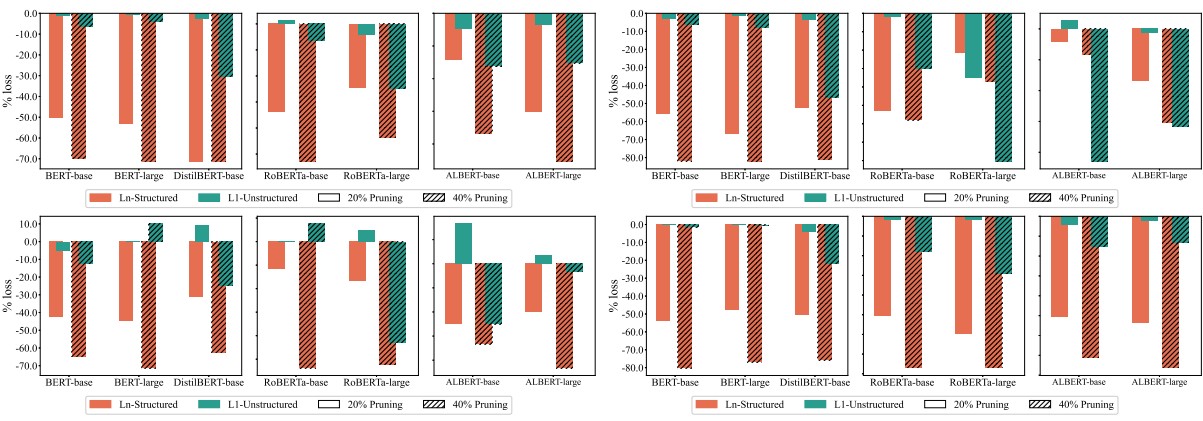

Figure 26: Drop in Top-1 accuracy across various datasets for encoder-only models for $FL_P$. Left-to-Right, Top-to-Bottom: TRex, Google-RE, SQUAD, ConceptNet (§4.4)

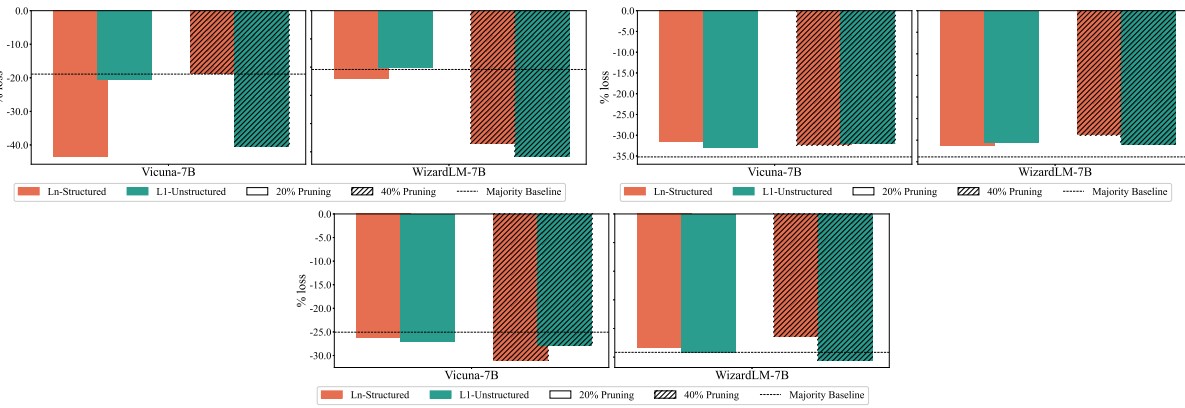

Figure 27: Drop in accuracy across various datasets for decoder-only models for $FL_P$. Left-to-Right, Top-to-Bottom: BoolQ, PIQA, and Winogrande (§4.4)

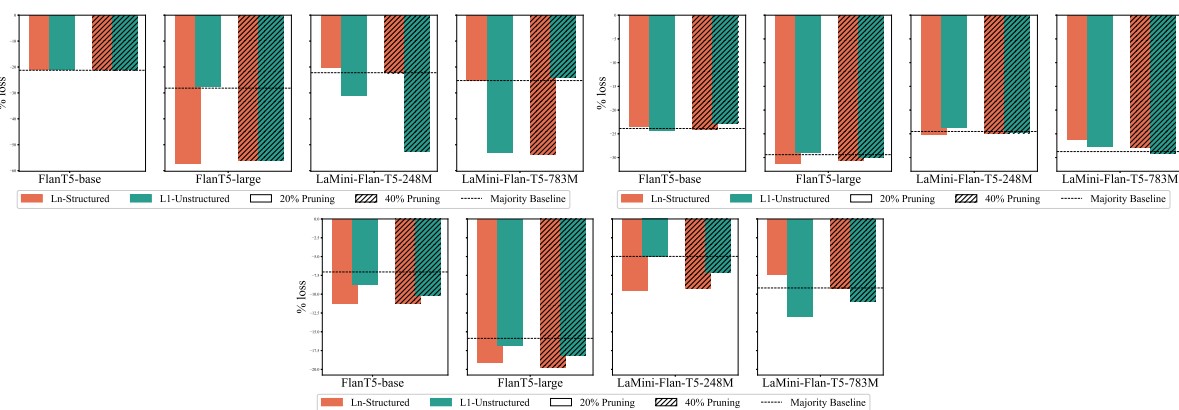

Figure 28: Drop in accuracy across various datasets for encoder-decoder models for $FL_P$. Left-to-Right: BoolQ, PIQA, and Winogrande (§4.4)