# OpenReview forum: "The Cost of Compression: Investigating the Impact of Compression on Parametric Knowledge in Language Models"
_EMNLP/2023/Conference — EMNLP 2023 Findings_

### Official Review · Reviewer_E2eZ · 2023-07-19

**Soundness:** 3

**Excitement:**

2: Mediocre: This paper makes marginal contributions (vs non-contemporaneous work), so I would rather not see it in the conference.

**Paper Topic And Main Contributions:**

The paper introduces a large experimental study on two compression techniques for LLM: pruning and quantization.
The authors focus on structured (L1) and unstructured (Lp) pruning for either attention modules, feed-forward modules, or the final dense layer.
For quantization, the authors focus on Post Training Dynamic Quantization (PTDQ), for either attention modules, feed-forward modules, or the final dense layer.

The authors conduct experiments on the LLM architectures:
Encoder-only models: BERT, RoBERTa, DistilBERT, and ALBERT.
Encoder-decoder models: Flan-T5, Lamini-Flan-T5.
Decoder-only models: Vicuna-7B, WizardLM-7B.

The authors reach several important findings within their comprehensive analysis.


**Reasons To Accept:**

The authors presented a comprehensive experiment on LLM compression varying in the methodologies and model architectures.
The reading was smooth and the paper is constructed elegantly. For example, I liked the fact that they included their main findings in textual boxes in order to ease the reading.

While most of the findings were quite intuitive and easy to understand, some of them were interesting to read, especially that pruning the attention modules for encoder-decoder models has a stronger impact than feed-forward modules in some cases, or that for encoder-only models, L1-unstructured leads to a smaller decrease in performance than L1-structured.

Besides that, I can't really point out any major contribution to literature.

**Reasons To Reject:**

As I mentioned above, most of the findings were not exciting and included statements such as:
- pruning FF modules has more effect than pruning attention modules since the FF module contains more parameters
- pruning + quantization of both FF and attention modules is worse than the compression of only one of these modules.
etc.

Besides this paper, there are already various papers that analyze the importance of compression techniques on LLM, and my feeling is that this paper does not substantially contributes to the work of others.

In my opinion, a much more interesting analysis will be to investigate the importance of specific layers/neurons (e.g.,  bottom layers vs. middle and top layers, attentions that are responsible for X vs. attention heads that are responsible for Y) or the importance of different compression techniques given the specific type of data domain (and not just showing average results across all tasks as the authors chose to do).

Although the authors supplied an explanation as to why choosing only one type of technique per method, I would like to see experiments on more pruning and quantization techniques as there are numerous of them. This fact really harms the generalizability of the reported findings.

**Reproducibility:**

3: Could reproduce the results with some difficulty. The settings of parameters are underspecified or subjectively determined; the training/evaluation data are not widely available.

**Reviewer Confidence:**

4: Quite sure. I tried to check the important points carefully. It's unlikely, though conceivable, that I missed something that should affect my ratings.

**Typos Grammar Style And Presentation Improvements:**

in line 70 the authors stated, "The applicability and effectiveness of such techniques for large ENCODER and ENCODER-DECODER models (Chung et al., 2022) has yet to be extensively studied."

This is certainly not the case, especially for encoder-only models where there is a vast amount of research that discusses the classification impact of many compression techniques.

---

> ### Author Rebuttal · Authors · 2023-08-27
>
> ### Pruning and FF vs. Attention Modules:
>
> While seemingly intuitive, we have found that the notion that pruning FF modules has a more significant effect than pruning attention modules (due to FF modules containing more parameters), **does not always occur**. Our findings indicate that the outcome depends on the model class and the compression percentage, as elaborated in Section 4.1. The overparameterization of FF modules in certain model families, such as encoder-decoder and decoder-only models, might shed light on this behavior, as mentioned in lines 429-433.
>
> Interestingly, even though FF modules have more parameters in each model we studied, there appears to be a consistent ratio (Appendix, Table 2) between attention and feed-forward parameters:
>
> - Encoder-only: Attention to Feed-Forward ratio of 1:3 for BERT, RoBERTa, and 1:2 for ALBERT, DistilBERT
> - Encoder-decoder: Attention to Feed-Forward ratio of 1:1
> - Decoder-only: Attention to Feed-Forward ratio of 1:2
>
> This pattern could account for some of our observations, where although FF modules contain more parameters, compressing them might not significantly impair performance in specific scenarios.
>
> ### Pruning + Quantization vs. Only One Module:
>
> We agree that the combined pruning and quantization of both FF and attention modules can yield worse results compared to the compression of only one of these modules. This is an intuitive insight **but nevertheless crucial**, since practitioners will likely rely on it. We have ensured that our findings align with logical expectations and have verified that they hold across various settings. Fortunately, our observations support these expectations.
>
> ### Substantial Contribution:
>
> We believe our contributions are substantial; these include:
>
> - Introducing an alternative metric for understanding LLM knowledge instead of the commonly-used metrics like BLEU score and perplexity.
> - Conducting a comprehensive study involving different model classes and various compression techniques, an area that has not been thoroughly explored before.
> - Modularizing LLMs to analyze the impact of different compression techniques on specific modules. This approach enhances our understanding of the significance of various blocks within an LLM and the consequences of compressing these blocks.
>
> ### Analyzing Individual Layers:
>
> For the BERT-base experiments with 20% pruning using l1-unstructured, we found that compressing final layers leads to a greater performance drop compared to initial layers. This discovery motivated us to conduct a separate analysis on final-layer compression, as described in Section 4.4. Our results align with similar findings from other research papers like [Mitchell, Eric, (2021)](https://arxiv.org/abs/2110.11309), [Mitchell, Eric, (2022)](https://proceedings.mlr.press/v162/mitchell22a.html), and [Meng, Kevin, (2022)](https://arxiv.org/abs/2202.05262). Importantly, we did not explore a bottom vs. middle/top layers analysis due to computational feasibility constraints for very large models (such as Vicuna-7B).
>
> |  Layer-num  | TRex | Google-Re | ConceptNet | Squad |
> |----|------|-----------|------------|-------|
> | 0 | 30.282 | 10.272 | 16.365 | 12.662 |
> | 1 | 30.285 | 10.29 | 16.347 | 13.312 |
> | 2 | 30.272 | 10.327 | 16.374 | 12.987 |
> | 3 | 30.18 | 10.272 | 16.321 | 13.312 |
> | 4 | 30.269 | 10.309 | 16.4 | 12.987 |
> | 5 | 30.199 | 10.272 | 16.347 | 12.662 |
> | 6 | 30.297 | 10.327 | 16.374 | 12.987 |
> | 7 | 30.307 | 10.309 | 16.383 | 12.987 |
> | 8 | 30.228 | 10.29 | 16.339 | 12.662 |
> | 9 | 30.297 | 10.29 | 16.4 | 13.312 |
> | 10 | 30.301 | 10.236 | 16.303 | 12.987 |
> | 11 | 30.326 | 10.2 | 16.374 | 12.987 |
> | 12 | 30.065 | 10.272 | 16.374 | 12.662 |
> | 13 | 29.903 | 10 | 16.392 | 12.338 |
>
>
> ### Representation of Results as Average Across All Tasks:
>
> We acknowledge that **in general**, averaging across different tasks might not be the optimal way to represent our results. However, **for the selected individual tasks**, there is no significant change in the observed patterns. Therefore, we opted to present average results in the main paper and included results for individual datasets in the appendix. This approach is similar to the approach taken in [Fig. 1](https://arxiv.org/abs/2208.07339).
>
> ### Experiments on More Pruning and Quantization Techniques:
>
> We appreciate the reviewer's suggestion regarding the generalizability of our findings to other compression techniques. We do acknowledge this limitation in our study. The reason we included these techniques is their standard nature---and the **critical gap in the literature** concerning comprehensive modular compression studies for various standard techniques across different model families. Our intent was to **fill this gap and initiate a direction** for understanding the effects of modular compression on model families.
>
> ### Reference for Encoder-only Models:
>
> Thank you for pointing out the need for a reference regarding the compression of encoder-only models. We have updated our draft accordingly. While there is a considerable body of research in the field of encoder-only model compression (as mentioned in lines 537-538), the recent compression techniques we explored (LLM-int8, SpQR, GPTQ) have not been extensively studied on large encoder-only models.

---

### Official Review · Reviewer_xMEC · 2023-07-28

**Soundness:** 2

**Excitement:**

3: Ambivalent: It has merits (e.g., it reports state-of-the-art results, the idea is nice), but there are key weaknesses (e.g., it describes incremental work), and it can significantly benefit from another round of revision. However, I won't object to accepting it if my co-reviewers champion it.

**Missing References:**

Perhaps highlight recent effort on direct distillation from LLMs https://arxiv.org/abs/2305.14864

**Paper Topic And Main Contributions:**

This paper presents a thorough decomposition of different pruning and quantization techniques for making LLMs more efficient. This work considers different ways to remove layers, connections or numerical precision for LMs ranging from BERT scale to larger >7B parameter models. The contribution of this paper is to study if different granularities of compression can help or hinder extrinsic performance. Each technique is described sufficiently and then different ablations / styles of compression are described and contrasted between.

The authors propose a series of Findings which may assist future researchers in developing a "rule of thumb" for evaluating how to shrink an LLM.

> I have read the authors rebuttal and acknowledge their additional details to their contributions. I'm concerned that the approach to statistical rigour has been "run many times for consistency" and not "run the Kolmogorov–Smirnov / Wilcoxon / Sign test to validate how different populations of models perform". Statistical rigour here means rejecting the null hypothesis of your outcomes, not consistency between random seeds.

**Questions For The Authors:**

* The contemplation of distillation is somewhat of an afterthought. I would query if this was a last minute addition. I would personally be in favor of removing references to this understudied technique to report stronger findings on better-scoped set of experiments.

**Reasons To Accept:**

* This paper presents a solid and thoughtful study on different methods of pruning or quantization for LLMs. This work will be helpful in community discussion of efficient methods from LLMs.

* The presentation of Findings in this paper is an exemplar for the *CL community. Everyone should follow their lead.

* Ideas are clear and well presented. This paper will help researchers gain an intuition on methods for compression (but see below for how to maybe improve this).

**Reasons To Reject:**

* The core text is lacking in quantitative analysis. Conclusions are drawn based on numbers being "higher" or "lower" than others with minimal contemplation of (a) scale of similarity or (b) statistical validity. The only quantitative mention in figure axes which are sometimes missing or carelessly annotated.
 The authors could improve this work by testing the statistical significance of their claims and analyzing if each compression technique has a *significant* change on performance. While I want to believe the claims made in the "Finding" box, and the presentation is convincing, it would be better received if approached with more statistical rigour.

* Conclusions and recommendations could be better summarised as "best practices" for future researchers. At present each finding is largely disconnected from each other. This makes a summary judgement from a lay researcher difficult to extract.

**Reproducibility:**

3: Could reproduce the results with some difficulty. The settings of parameters are underspecified or subjectively determined; the training/evaluation data are not widely available.

**Reviewer Confidence:**

3: Pretty sure, but there's a chance I missed something. Although I have a good feel for this area in general, I did not carefully check the paper's details, e.g., the math, experimental design, or novelty.

**Typos Grammar Style And Presentation Improvements:**

* Recommend rendering your figures in .eps format and directly rendering in LaTeX for better readability, no text warping and a consistent aspect ratio

* Fig 6 - 10 would benefit from bar labels with the actual numbers. Similar for any bar chart in the appendix.

* Recommend a clearer split between main text figures and appendix figures. At line 451-452, please state that the Figures are not in the main text and can be found in the appendix. The paper must be airtight without the appendix which reviewers are not required to read.

---

> ### Author Rebuttal · Authors · 2023-08-27
>
> ### On statistical validity:
> Our experiments fall into **two categories**. Some are deterministic, such as pruning, where we used l1-unstructured pruning (see, for example, line 299 of the submitted draft). In this setting, there is no concept of statistical significance as the experiment results are consistent across multiple runs.
>
> In the second case, such as for our quantization experiments, there is an element of randomness. For quantization, this is due to our use of PTDQ, which computes a dynamic clipping range (described in line 310). We deliberately **struck a balance** between the number of trials per setting and the overall number of settings studied. Consequently, we ran experiments with multiple seeds and recorded confidence intervals, as demonstrated in the table below. This approach aligns with the standard procedure in large-scale machine learning, where numerous settings for large models need to be tested. **Importantly, we observed relatively minor variation** (standard deviation ranging from 1-5%). For this reason, we are confident in the meaningfulness of our findings.
>
> ### BERT-base Experiments on Squad Dataset:
> The results are as follows:
>
> |       | Overall | Attention only | Output only |
> |-------|---------|----------------|-------------|
> | Seed 40 | 5.844 | 13.312 | 3.896 |
> | Seed 50 | 5.844 | 13.312 | 5.195 |
> | Seed 60 | 6.169 | 12.987 | 4.221 |
> | **Mean** | 5.952 | 13.204 | 4.437 |
> | **Standard Deviation (sample)** | 0.188 | 0.188 | 0.676 |
> | **Standard Deviation (population)** | 0.153 | 0.153 | 0.676 |
>
> ### BERT-large Experiments on Squad Dataset:
> For this setting, we have:
>
> |       | Overall | Attention only | Output only |
> |-------|---------|----------------|-------------|
> | Seed 40 | 2.273 | 16.883 | 2.597 |
> | Seed 50 | 2.922 | 14.286 | 3.896 |
> | Seed 60 | 2.922 | 15.909 | 4.545 |
> | **Mean** | 2.706 | 15.693 | 3.679 |
> | **Standard Deviation (sample)** | 0.375 | 1.312 | 0.992 |
> | **Standard Deviation (population)** | 0.306 | 1.071 | 0.992 |
>
> ### On "best practices" section:
>
> Thank you for the feedback! We entirely agree with the reviewer's suggestions and will incorporate these changes in the updated version of our paper.
>
> ### On distillation experiments:
>
> In order to avoid potential variation due to chosen settings when distilling from scratch, we used standard pre-distilled models (such as distilBERT and Lamini-T5) in our study. Notably, **we found that distillation exhibited similar effects** on compression as seen with base models. This discovery suggests an intriguing possible approach: distilling a model before applying compression techniques like pruning and quantization.
>
> ### On formatting suggestions:
>
> Thank you for the formatting suggestions. We appreciate the input and have already made the necessary adjustments to our updated draft.

---

### Official Review · Reviewer_syfP · 2023-08-03

**Soundness:** 3

**Excitement:**

2: Mediocre: This paper makes marginal contributions (vs non-contemporaneous work), so I would rather not see it in the conference.

**Paper Topic And Main Contributions:**

The main contribution made by this work is an experimental study and evaluation protocol to investigate the loss of knowledge when compressing the learned weights of mainstream pre-trained LMs. Specifically, it studies models of various structures, i.e., encode-only, encoder-decoder, and decoder-only.  Compression on different modules (FF and attention) are also done with interesting findings.

**Questions For The Authors:**

1. The models being compared in different categories vary largely in terms of model sizes. Decoder-only models are considerable larger than those of encoder-decoder and encoder-only. It is likely that the behaviour is different with different model sizes. Had you verified if the same findings hold with smaller decoder-only models? or with larger encoder-decoder models such as flan-T5-XXL?

2. As being pointed out, attention contains more parameters than FF in some cases. It is not very informative to only look at the pruning levels. Would it be more meaningful to also plot the performance drop against compression ratio?

3. For encoder-decoder models, self-attention and cross-attention might have different functions, and thus, it might be more favourable to study them separately. Also, attentions in encoder and in decoder should be examined in the same way.

4. For the hypotheses made in Line 433-436, can it be verified by reducing the number of parameters in FF layers (e.g., by using a low rank approximation)?

**Reasons To Accept:**

This paper focused on the preservation of knowledge which has been somehow overlooked in existing study of compressing LLMs. Experiments are done with many mainstream LLMs, and the writing is  overall easy to follow. Compression methods for LLMs might potentially benefit from the findings made in this paper.

**Reasons To Reject:**

Findings seem a bit superficial. The author has provided a set of hypotheses, but few of them have been verified or examined. It could be more insightful with additional experiments. I will elaborate more on the my doubts in the following section.

**Reproducibility:**

4: Could mostly reproduce the results, but there may be some variation because of sample variance or minor variations in their interpretation of the protocol or method.

**Reviewer Confidence:**

4: Quite sure. I tried to check the important points carefully. It's unlikely, though conceivable, that I missed something that should affect my ratings.

---

> ### Author Rebuttal · Authors · 2023-08-27
>
> ### On comparison with other sized models:
>
> Thank you! We have observed similar findings when examining larger encoder-decoder models, specifically the FlanT5-XL with 3B parameters. Our observation highlights that compressing Attention (Att) modules results in a more significant drop compared to Feed-Forward (FF) modules. The rationale behind this observation is discussed in lines 428-433 of our work.
>
> #### Attention GP:
>
> |  | Baseline | 10% | 20% | 30% | 40% | 50% | 60% | 70% | 80% | 90% |
> |---------|----|----|----|----|----|----|----|----|----|----|
> | BoolQ   | 0.8945 | 0.8633 | 0.8584 | 0.8339 | 0.6073 | 0.4682 | 0.3838 | 0.3786 | 0.382 | 0.3789
> | PIQA    | 0.7568 | 0.7584 | 0.753 | 0.7437 | 0.6915 | 0.6376 | 0.5914 | 0.5571 | 0.5457 | 0.537
> | Winogrande | 0.693 | 0.6953 | 0.6961 | 0.674 | 0.5446 | 0.4996 | 0.4988 | 0.5264 | 0.4964 | 0.5012
>
> #### Feed-Forward GP:
>
> | | Baseline | 10% | 20% | 30% | 40% | 50% | 60% | 70% | 80% | 90% |
> |---------|----|----|----|----|----|----|----|----|----|----|
> | BoolQ   | 0.8945 | 0.8642 | 0.8618 | 0.8596 | 0.8498 | 0.8428 | 0.8214 | 0.5486 | 0.3789 | 0.481
> | PIQA    | 0.7568 | 0.7557 | 0.7557 | 0.7546 | 0.7552 | 0.7367 | 0.7116 | 0.6045 | 0.518 | 0.5201
> | Winogrande | 0.693 | 0.6953 | 0.7032 | 0.6851 | 0.6961 | 0.6867 | 0.6361 | 0.5201 | 0.5004 | 0.4822
>
> Note: Majority baselines are - BoolQ: 0.621, PIQA: 0.504, Winogrande: 0.504
>
> Critically, our study focuses on **analyzing individual model families independently** rather than comparing encoder-only, encoder-decoder, and decoder-only models. This approach is reflected in the way we present our findings.
>
> ### On plotting the performance against compression ratio:
>
> Thank you! We have generated plots for all three model families, **yielding insights consistent with the discussion** in lines 428-433. The plots visually resemble those presented in our paper, albeit with a skewed $x$-axis. Consequently, given a specific performance drop percentage, it is highly likely that we can achieve greater parameter compression by targeting FF modules rather than Att modules. It is worth noting that across all the models studied, FF modules have more parameters than attention modules.
>
> Our plot construction is straightforward; we describe it as follows (line 286) in the submitted draft: "We note that the number of parameters compressed differs for different settings. We record all of the values required for normalizing measurements. However, our focus is predominantly aimed at understanding the effects of compressing modules and their combinations rather than presenting normalized results, and our insights reflect this framing. We provide full parameter counts that permit normalized quantities that can be used by practitioners who seek to directly apply our work."
>
> ### On studying attentions separately:
>
> Thank you for the feedback! We acknowledge the potential divergence in the functions of self and cross attention, as well as their roles within encoder and decoder modules. We will address this aspect in the updated draft. We are conducting additional experiments to enable doing so. Additionally, our analysis in Figure 8 delves into the combined impacts of pruning and quantization on different types of attention modules.
>
> ### On low-rank approximation to verify claims in lines 433-436:
>
> We appreciate your suggestion! We will certainly incorporate the investigation of low-rank approximation in our updated draft through supplementary experiments. We also believe that a dedicated study on addressing the over-parametrization of LLM modules is a great topic for a future paper, and we plan to pursue this goal.

---

### Meta-Review · Area_Chair_g8q5 · 2023-09-19

**Recommendation:** 3

**Metareview:**

This work investigates the effect of compressions such as pruning or quantization on the parametric knowledge learned by different types and sizes of pre-trained language models.

As the reviewer mentioned, the paper presents a solid study on compression of LLMs by focusing on the preservation of knowledge which has been somehow overlooked in existing studies in the compression context. Furthermore, the findings are helpful for the researchers and practitioners for both further compression studies and the use of such methods. Finally, the presentation of the findings demonstrates a nice example and makes the paper easier to follow.

However, the reviewers also noted that the paper has limited contribution to the existing literature. An important part of the paper focuses attention and the feedforward modules. More fine-grained analysis such as layers and attention heads are suggested. Furthermore, the issues such as varying model size in different model families (note that as mentioned in LLM-int8 some LLMs larger than 6B face a stronger quantization challenge), and separate analyses for different attention modules remain to be investigated within the context of this paper.

---

### Decision · Program_Chairs · 2023-10-07

**Decision:**

Accept-Findings

**Comment:**

This work investigates the effect of compressions such as pruning or quantization on the parametric knowledge learned by different types and sizes of pre-trained language models.

As the reviewer mentioned, the paper presents a solid study on compression of LLMs by focusing on the preservation of knowledge which has been somehow overlooked in existing studies in the compression context. Furthermore, the findings are helpful for the researchers and practitioners for both further compression studies and the use of such methods. Finally, the presentation of the findings demonstrates a nice example and makes the paper easier to follow.

However, the reviewers also noted that the paper has limited contribution to the existing literature. An important part of the paper focuses attention and the feedforward modules. More fine-grained analysis such as layers and attention heads are suggested. Furthermore, the issues such as varying model size in different model families (note that as mentioned in LLM-int8 some LLMs larger than 6B face a stronger quantization challenge), and separate analyses for different attention modules remain to be investigated within the context of this paper.